# β-Cell Maturation and Identity in Health and Disease

**DOI:** 10.3390/ijms20215417

**Published:** 2019-10-30

**Authors:** Ciro Salinno, Perla Cota, Aimée Bastidas-Ponce, Marta Tarquis-Medina, Heiko Lickert, Mostafa Bakhti

**Affiliations:** 1Institute of Diabetes and Regeneration Research, Helmholtz Zentrum München, D-85764 Neuherberg, Germany; ciro.salinno@helmholtz-muenchen.de (C.S.); perla.cota@helmholtz-muenchen.de (P.C.); aimee.bastidas-ponce@helmholtz-muenchen.de (A.B.-P.); marta.medina@helmholtz-muenchen.de (M.T.-M.); 2German Center for Diabetes Research (DZD), D-85764 Neuherberg, Germany; 3Institute of Stem Cell Research, Helmholtz Zentrum München, D-85764 Neuherberg, Germany; 4School of Medicine, Technical University of Munich, 81675 Munich, Germany

**Keywords:** β-cell, maturation, postnatal, identity, dysfunction, dedifferentiation, transdifferentiation, senescence, SC-β-cells, diabetes

## Abstract

The exponential increase of patients with diabetes mellitus urges for novel therapeutic strategies to reduce the socioeconomic burden of this disease. The loss or dysfunction of insulin-producing β-cells, in patients with type 1 and type 2 diabetes respectively, put these cells at the center of the disease initiation and progression. Therefore, major efforts have been taken to restore the β-cell mass by cell-replacement or regeneration approaches. Implementing novel therapies requires deciphering the developmental mechanisms that generate β-cells and determine the acquisition of their physiological phenotype. In this review, we summarize the current understanding of the mechanisms that coordinate the postnatal maturation of β-cells and define their functional identity. Furthermore, we discuss different routes by which β-cells lose their features and functionality in type 1 and 2 diabetic conditions. We then focus on potential mechanisms to restore the functionality of those β-cell populations that have lost their functional phenotype. Finally, we discuss the recent progress and remaining challenges facing the generation of functional mature β-cells from stem cells for cell-replacement therapy for diabetes treatment.

## 1. Introduction

Diabetes mellitus (DM) is a chronic condition characterized by impairment of blood glucose homeostasis, resulting in hyperglycemia and a series of secondary complications, such as cardiopathy, neuropathy, nephropathy, and retinopathy. There are two main forms of diabetes. Type 1 diabetes (T1D) is an autoimmune disease where insulin-producing pancreatic beta cells (β-cells) are destroyed by the immune system. In contrast, insulin resistance and progressive dysfunction of β-cells characterize type 2 diabetes (T2D) [1,2]. Current treatments are only able to ameliorate diabetes symptoms by decreasing/normalizing the blood glucose levels without halting the causes of the disease. Administration of insulin remains the most common treatment for patients with T1D and the last treatment option for patients with T2D. However, insulin treatment is associated with some risk of hypoglycemic episodes, weight gain and increased incidence of cancer [3]. The only curative approaches are bariatric surgery for T2D [4] and transplantation of pancreatic islets of Langerhans from cadaveric donors, especially for T1D [5]. Unfortunately, this last approach is neither easily applicable nor permanent. First, the shortage of donor organs makes the transplantation option exclusively available to patients that fulfill a strict severity criterion. Second, the patients that receive the cadaveric islets are at risk of (auto-)immune-rejection, thus they are treated with immunosuppressive drugs, with an increased associated risk of infections and cancer [6]. Considering T2D as a possible reversible disease, alternative therapeutic strategies are being developed. Removing the main causes of the diabetic condition is at the base of the newest approaches, for example by removing glucotoxicity, one of the main driver of β-cell loss and identity in T2D [7]. Thus, it is possible to improve β-cell function by reestablishing cellular maturation and identity and to protect and regenerate dysfunctional β-cells during disease progression.

Theoretically, regenerative approaches are another alternative option for diabetes treatment that includes: (i) reestablishing or enhancing the healthy cellular phenotype and (ii) replacing the lost and/or dysfunctional cells. The first strategy focuses on finding drugs and small molecules (a) to restore the physiological signaling pathways lost in disease [8,9,10], (b) to remove the dysfunctional cells from the islets [11,12], or (c) ameliorate the micro-environmental conditions that sustain the impairment of β-cells [13,14,15,16]. The goal of this approach is to redirect the dysfunctional β-cells towards a healthy functional state. However, to target β-cells or specifically deliver drugs to these cells remains a major obstacle. The second strategy focuses on the screening of compounds to trigger β-cell neogenesis, transdifferentiation of non-β islet cells towards β-cells or the endogenous expansion of existing β-cells [17,18,19,20]. Alternatively, great efforts are put together to replace β-cells by using stem cell derived β-cells (SC-β-cells) as source for transplantation [21,22,23,24,25,26,27]. These SC-β-cells should be fully functional similar to the endogenous mature adult β-cells, in order to be used for the clinical settings. So far, no differentiation protocol has achieved the generation of fully functional mature β-cells that present comparable glucose-stimulated insulin secretion (GSIS) to human adult islets. Over the past two decades, groundbreaking research has been carried on to decipher β-cell maturation process. These cells arise during embryonic development [28,29,30] with an immature phenotype [31,32,33]. After birth, a sequence of molecular and metabolic changes lead to β-cell maturation, which enables these cells to respond with an appropriate insulin release to fluctuating glucose levels. To fulfill their physiological function, β-cells actively preserve this maturation machinery that defines their functional identity. Numerous reports have shown the loss of β-cell maturation and identity in diabetic conditions [34,35,36]. Therefore, it is essential to understand the maturation process in detail in order to prevent the loss of maturity or restoring maturation state of those β-cell that lost their identity.

Recent studies have shown that not all β-cells acquire maturation at the same time. Furthermore, mature β-cells represent heterogonous populations in terms of phenotype and functionality. As there are excellent reviews on β-cell heterogeneity [37,38,39,40,41,42], here we focus on the current understanding of the mechanisms that regulate β-cell maturation and identity, in healthy and diabetic conditions. First, we summarize the characteristics (markers, functionality, and signaling pathways) that allow distinguishing immature and mature β-cells. Second, we provide an overview of what is currently known about the loss of β-cell identity and the three main phenotypes that have been identified in diabetic conditions: dedifferentiated, senescence, and transdifferentiation of β-cells. We further summarize the state of the art of therapeutic approaches to reestablish the mature phenotype or to replenish the β-cell mass. Finally, we provide a short overview of the most recent advancements regarding the differentiation protocols to generate functional mature SC-β-cells in vitro. A better understanding of the β-cell differentiation and maturation as well as deciphering the dysfunctional β-cell states arising in disease can lead future research to develop better therapeutic strategies and even permanent disease modification for patients with diabetes.

## 2. Embryonic Development: Rise of β-Cells

In rodents, endocrine cells are formed at two stages or transitions during embryonic development. At the primary transition (E9.0–E12.5) the main differentiated endocrine cells are α-cells. In contrast, during secondary transition (E12.5–E15.5) all other endocrine cell types including β-cells are formed [32,33,43,44]. Notably, in humans, it is thought that only the secondary transition exists and that gives rise to all hormone-secreting endocrine cells [45]. In rodents and humans, endocrinogenesis initiates with the emergence of the mitotic endocrine progenitors that are marked by the expression of low levels (Ngn3^low^) of the transcription factor (TF) neurogenin 3 (Neurog3, Ngn3). The increase in expression levels of Ngn3 results in generation of low-cycling endocrine precursors (Ngn3^high^) [46,47]. Then the endocrine precursors generate E26 transformation-specific transcriptional factor (Fev)-expressing cells, which do not yet express hormones [48]. Further differentiation of Fev^+^ cells results in the acquisition of secretory machinery components and hormones to form differentiated endocrine cells [28,48]. How the different endocrine cell types are generated from a single progenitor pool is not well understood yet. The status of pancreatic epithelium and developmental timing of endocrine differentiation are among possible mechanisms [28,49,50]. Moreover, it has been suggested that endocrine precursors are unipotent and cell fate commitment towards specific endocrine subtype is already determined at this stage [46,51]. More evidence comes from a recent study showing that endocrine precursors are transcriptionally heterogeneous and they differentially express a set of specific signature genes that might define their fate towards α- or β-cells [28]. However, most of these signature genes are poorly characterized and future studies should uncover their possible function during endocrine lineage allocation. Recent studies showed how mouse and human pancreas follow markedly different paths of development [52,53]. In conclusion, further studies are required to define the expression of these signature genes in humans and whether they have evolutionary conserved function during endocrinogenesis.

Endocrine differentiation is tightly regulated by dynamic changes in gene regulatory networks (Figure 1). The master regulator of endocrine progenitor-precursor (EP) formation and differentiation is Ngn3. This TF is transiently expressed in EPs and its function is indispensable for endocrine cell formation [54,55]. As differentiation proceeds, the generation of different endocrine cell types relies on the expression of distinct specific TFs. For instance, higher expression levels of Arx and Pax4 favor formation of α- and β/δ-cells, respectively [56]. Moreover, differentiation toward β-cell fate depends on the expression of several other TFs, such as Foxa2, Pdx1, Nkx6-1, Neurod1, Nkx2-2 and Mnx1 [57,58,59,60]. Although most of these TFs have been shown to be expressed during human β-cell formation [45,61], their expression pattern and function have not been studied as detailed as in rodents. In contrast to rodents, NKX2-2 is only expressed in human endocrine cells but not in pancreatic progenitors. Furthermore, GATA6 is necessary only for the formation of the human but not rodent pancreas [45,62]. Additionally, human β-cells co-express MAFA and MAFB, though MAFA is expressed in juvenile human β-cells, and there is no consensus about its expression in SC-β-cells [43,63,64]. Although these studies have shed light on the expression pattern of some TFs during human endocrinogenesis, deeper analyses are required to uncover the similarities and differences between mouse and human endocrine cell formation. Combining ex vivo modeling systems such as 3D cyst or organoids [30,65] with single-cell genomic and transcriptomic profiling will help to provide a more comprehensive picture of gene regulatory changes during human endocrinogenesis.

## 3. Postnatal Development: Road to Maturity

Most of our current understanding of β-cell maturation is derived from rodent studies. In mouse, embryonic β-cells are described as immature, highly proliferative and unipotent albeit still plastic. Although at this stage β-cells present insulin granules and display high basal insulin, until which extent they can secret insulin is not well characterized [31,32]. After birth, β-cells are abundant and organized in clusters, so called proto-islets [33,66]. However, at this stage these cells do not yet acquire a mature phenotype to properly secret insulin in response to glucose levels [67]. The events that conclude the maturation process occur in postnatal stages. β-cells follow a biphasic pattern of maturation [68], possibly adapting to the neonatal type of diet. The first two weeks after birth define a first maturation wave [31]. Here, it is possible to observe a general increase of the endocrine mass, implying that β-cells are still proliferative, characteristic that is progressively lost [69]. During this stage, β-cells acquire all the TFs and machinery necessary for the establishment of the adult β-cell identity (Figure 1). The second wave of maturation coincides with the third week of life and the weaning period [70]. During this time, β-cells differentially regulate metabolic pathways defining the mature functional landscape [11]. For simplicity, we describe and divide the postnatal maturation process according to three aspects: (i) markers, (ii) functionality, and (iii) signaling pathways. Clearly, the three aspects are tightly connected to one another, contributing to the complex identity of mature β-cells.

### 3.1. Markers

At birth, all β-cells possess signature genes, which are important for the establishment and maintenance of β-cell identity: Nkx2-2, Pdx1, Nkx6-1, and Neurod1 are some examples. In the past years, several markers have been characterized in order to identify sub-clusters of β-cells during development, healthy or diseased states (Figure 2). Urocortin3 (UCN3) is a secreted peptide that is expressed at low levels in β-cells already before birth. Within the following 2 weeks, almost all β-cells possess high levels of this peptide [31]. Thus, the acquisition of Ucn3 marks the early phase of postnatal maturation. The role of Ucn3 is still not completely known. One study linked Ucn3 activity with somatostatin-dependent negative feedback of insulin secretion [71]. Of note, UCN3 is expressed, in humans, in both α- and β-cells and in human embryonic stem cell (hESC)-derived endocrine cells upon transplantation [72]. Similarly, a member of Synaptotagmin (Syt) family, Syt4, is upregulated during the postnatal stages. The role of this protein is fundamental for the reduction of calcium sensitivity, high in immature β-cells, that directly regulates the exocytosis of insulin granules [73]. Contrary to Ucn3 and Syt4, Neuropeptide Y (NPY) marks immature β-cells in both mouse and human [74]. In mouse, the levels of Npy drop dramatically in the first two weeks of life, and is barely detectable at one month, overlapping with the first phase of β-cell maturation and acquisition of functionality. The role of this neuropeptide is still not fully understood. Some evidence suggests that it might contribute to the maintenance of the immature phenotype, promoting proliferation and high basal insulin secretion [74,75].

Among TFs, the Maf family is important for the establishment of β-cell identity and functionality [76]. In rodents, MafB is highly expressed in embryonic β-cells and is rapidly downregulated after birth. During this period, MafA progressively substitutes MafB and regulates the expression of genes, such as Glut2, GLP-1 receptor, prohormone convertase-1/3, and pyruvate carboxylase that characterize glucose sensing and insulin secretion [77,78,79]. Thus, the switch from MafB (which remains expressed in α-cells) to MafA is crucial and characterizes the late phase of postnatal maturation [76,80]. Different from rodents, in humans, MAFB remains highly expressed in adult β-cells [81,82]. Along this line, MAFB knockdown results in dysregulation of β-cell transcriptional program, providing an important proof of non-redundancy function of the two TFs in humans [83]. Future studies should address this peculiarity of human β-cells, in order to understand the mechanisms that sustain MAFB function.

Another marker recently characterized is Flattop (Fltp), a Wnt/Planar Cell Polarity (PCP) effector protein [84]. This marker has been exploited to distinguish an immature pool (Fltp^−^) and terminally mature (Fltp^+^) β-cells. The Fltp^−^ β-cells are characterized by a higher proliferation rate (both in homeostasis and pregnancy) and differential mRNA expression of GPCR, Wnt and MAPK signaling components. The Fltp^+^ cells feature a higher expression of β-cell functional genes (i.e., Slc2a2, Nkx6-1, Ucn3, MafA, etc.), an increased number of mature secretory granules, enhanced mitochondrial physiology and higher static GSIS [85]. Thus, the study has linked Wnt/PCP activity to cell cycle exit and therefore acquisition of functional maturity. Similarly, in humans, the surface markers ST8SIA1 and CD9 distinguish four antigenically subgroups of β-cells in adult islets. These groups differ in gene expression profiles and functionally. Although all groups express comparable levels of INS, PDX1, NKX6-1, and MAFA, they differentially express more than 200 genes including GLUT2, PPP1R1A, SUR2, and G6PC2 that are involved insulin secretion. These differences are reflected in their functionality since the group of β-cells lacking both surface markers show the lowest basal insulin secretion and the highest GSIS. Finally, the ratio of the four subpopulations changes in diabetes. It is of great interest to understand what are the determining mechanisms that drive this change [86]. Of note, the heterogeneity of β-cells seems to be an intrinsic and thus relevant feature. This topic has been broadly described elsewhere and therefore, we have not extensively discussed it in this review [37,38,40,42]. Altogether, an increased number of studies have adopted a series of novel markers to distinguish β-cell subpopulations, underlying different physiological attributes. Understanding the dynamics of acquisition and maintenance of these markers is a powerful tool to assess the pathophysiological state of β-cells, in health and diseases, and to determine the efficacy of therapeutic treatments for diabetes.

### 3.2. Functionality

The presence or absence of certain markers is associated with functional characteristics that distinguish immature from mature β-cells. However, not all β-cells develop at the same time and some remain with distinct immature characteristics during adult life. This heterogeneity mostly develops during the perinatal age and it is maintained, albeit debunked, in adulthood. Several studies [85,86,87,88] have demonstrated the existence of different types of β-cells within the islets. The ratio of the different subpopulations usually shifts under metabolic stress or diabetes progression. Therefore, the maintenance of the β-cell heterogeneity does not seem to be random but contextual. In this section, we describe the most prominent functional features of β-cells during maturation. The proliferative capacity of β-cells, both in mouse and human, is strictly confined to the early stages of life (for more information see reviews [89,90,91]) (Figure 2) or to high demanding physiological conditions (such as pregnancy [92]). This idea has been recently strengthened by isotope labeling experiments, in which the basal bodies of primary cilia of β-cells remain unchanged for long period of time, thus proving the absence of cell cycle [93]. On contrary, β-cell proliferation is increased upon insulin resistance in obesity [94]. The cell cycle exit coincides with a series of molecular changes in β-cells that lead to the terminal maturation process [95,96,97]. In healthy conditions, only a small pool of β-cells retain low proliferative capacity, which is linked to an immature functional phenotype [86]. Thus, the reduced proliferative capacity is one of the strongest features of immature β-cells. Understanding how to manipulate the cell cycle in these cells is foreseen as a possible therapeutic treatment option to restore the lost β-cells mass.

The glucose stimulated insulin secretion (GSIS) is the best functional characteristic of mature β-cells. The GSIS is defined as the capability of β-cells to secrete an appropriate amount of insulin in response to proportional extracellular glucose stimuli. The GSIS is the sum of a multitude of cellular processes [98,99], such as sensing of glucose, which enters β-cells via the specific glucose transporter (Glut2 for rodents, GLUT1 for humans). The expression of this transporter is one of the main functional characteristics that distinguish a mature (even if not fully mature) from an immature or diseased β-cells (Figure 2). After entering into β-cells, glucose is quickly converted into glucose-6-phosphate (G6P), which is redirected in the two main metabolic processes: glycolysis and oxidative phosphorylation. The events result in the production of ATP and other molecules that also serve as coupling factors. In this context, the mitochondria biogenesis and metabolisms play a crucial role in the development of the mature GSIS, since it has been proved that this organelle actively participates in the control and enhancement of insulin secretion. The final goal of the actions that take place in β-cells after the entrance of glucose is to induce the exocytosis of the cytoplasmic insulin granules. In order to do so, a pattern of electrical signals needs to be generated by the fast activation and deactivation of ions channels. The increase of the ATP/ADP ratio induces the closure of the K_ATP_ channels that generates a depolarization of the plasma membrane. This triggers the action potential that opens the voltage-gated Ca^2+^ channels. The influx of this ion has the fundamental role to induce the exocytosis of the insulin vesicles. Many other factors can influence insulin secretion, as the amino acids arginine and leucine, or the hormone GLP-1 (for more details see reviews: [100,101,102,103]). All of the above mention features are tightly connected to each other, creating an extremely complex molecular network that is developed during the postnatal maturation process.

Comparative studies among different ages showed that the biological processes beneath the GSIS are activated only during the postnatal development [104,105]. In fact, the neonatal islets are considered as “leaky”, meaning that β-cells secrete high insulin levels in response to low glucose stimuli (defined as high basal insulin secretion) [31,106]. Once acquired key components, such as Syt4, the exocytosis of insulin granules is less sensitive to low calcium levels, therefore β-cells secrete low insulin concentrations at low glucose levels and high insulin concentrations, at high glucose levels, in a biphasic fashion [107]. Recent studies addressed the topic of the functional heterogeneity in healthy adult islets, in zebrafish, mice, and humans. It has been shown that putative pacemaker β-cells, called hub or leader cells, are heavily connected with neighbor cells and through rhythmic calcium oscillations coordinate the electrophysiological response to glucose stimuli [87,108,109]. Thus, the presence of hub or leader cells has a major contribution to the collective regulatory behavior of islets in secreting insulin. Of interest is the understanding of when these hub cells arise and how this functional dominance is established.

### 3.3. Signaling Pathways

A complete understanding of the β-cell maturation process has to include the knowledge of the pathways that are differentially regulated (Figure 2). The mTOR (mammalian target of rapamycin) pathway orchestrates a plethora of cellular activities including proliferation and growth [110] in response to external stimuli such as nutrients or survival signals. This pathway is also the predominant signaling in charge of the early postnatal β-cell development. The beginning of oral feeding after birth, thus the transition from blood supplied nutrition type to an enteral one, induces the upregulation of the mTOR signaling pathway. According to Sinagoga and colleagues, the mTORC1 complex contributes to the progression of the functional maturation, while the mTORC2 branch regulates islet architecture and mass [111]. Other studies also confirmed that the mTORC1 signaling significantly affects the glucose metabolism [112] and therefore the advancements in the tuning of GSIS.

Another pathway that has recently gained attention in the context of β-cell maturation is AMPK (5′ AMP-activated protein kinase) signaling. The AMPK complex is a cellular energy and stress sensor, responding to the available intracellular levels of ATP [113]. This pathway is already the target of diabetes treatments and it is known to be involved in the regulation of β-cell mass and insulin secretion, among others [114,115]. Previous studies have shown how weaning, thus the switch from high-fat milk to high-carbohydrate food, triggers the β-cell maturation enhancing oxidative phosphorylation at high glucose levels while blunting it at basal glucose levels [70]. In a similar fashion, Jaafar and colleagues have demonstrated how the weaning process induces the activation of the AMPK signaling, which antagonizes and inhibits the mTORC1 complex. By continuous feeding of litters with milk-based gavages, longer after weaning age, they maintained neonatal levels of mTORC1 in β-cells. Thus, AMPK signaling is at the center of the second phase of maturation, inhibiting the mTORC1 pathway and enhancing mitochondrial biogenesis and oxidative metabolism [11]. On the same line, the ERRγ transcriptional network has been proved to induce the metabolic transition in the postnatal age that determines the acquisition of the mature β-cell functionality, among which the typical GSIS [116].

Wnt signaling pathway is known to be fundamental for the endocrinogenesis process during the embryonic development [117,118,119]. A recent study has also engaged the use of a Wnt/PCP effector protein, Fltp, to demonstrate the association between non-canonical Wnt signaling and the β-cell maturation phenotype [86]. However, the molecular mechanisms through which Wnt/PCP regulates the maturation process have not been deciphered yet. One possible scenario is that through rearrangement of cytoskeleton components and β-cell polarity [120], non-canonical Wnt signaling impacts islet compaction and architecture to determine a higher order of tissue organization, required for β-cell function [42,121]. As supporting evidence, the treatment of mouse islets and Min6 cells with non-canonical Wnt ligands, Wnt5a, increases the expression of maturation markers Ucn3, Nkx6-1 and Glut2. Furthermore, the treatment of EndoC-β H1 human cell line and human islets with the ligands Wnt4 and Wnt5a respectively upregulates JNK activity and improves GSIS [86]. Similarly, secreted frizzled-related protein-5 (Sfrp5) reduces the proliferation rate and improves the GSIS in Ins1 cells through upregulation of JNK [122]. It should be noted that the function of non-canonical Wnt signaling in the acquisition of the mature phenotype could partially be indirect, via inhibition of the cell cycle. Despite these studies, the remained challenge is to decipher the interplay between the abovementioned signaling pathways including mTOR, AMPK and Wnt signaling during β-cell maturation. Additionally, it is possible that other signaling pathways such as Hippo signaling also play a role in establishment of the mature β-cell phenotype [123]. Detailed analysis of signaling pathways involved in β-cell maturation might provide novel therapeutic targets to restore the function of β-cells in diabetes and the implementation of novel steps for the in vitro SC-β-cell differentiation.

## 4. Diabetic Conditions: β-Cells Derail

β-cell dysfunction is a hallmark of several metabolic disorders, such as diabetes and it is characterized by the lack of function and loss of identity [34,35,36]. Oxidative stress, high levels of glucose and lipids, inflammatory cytokines, and altered gene expression represent unhealthy environments and are only among the causes of β-cell dysfunction [124]. Moreover, β-cells appear to acquire a level of plasticity in response to particular stressful environments provided, in this case, by T1D or T2D. Here, we aim to provide an overview of the different β-cell subpopulations or states reported in disease: dedifferentiated, transdifferentiated and more recently senescent β-cells.

### 4.1. Dedifferentiated β-Cells

A very well described feature of β-cells during diabetes is their loss of functional mature identity (Figure 3) [125]. A number of diabetic mouse models and human samples have been used to identify the factors that force β-cells to lose their mature features under stressful environments (such as glucotoxicity and lipotoxicity) [126,127,128,129,130,131,132,133]. The term “dedifferentiation” has been assigned to the reported dysfunctional β-cells, based on the idea that these cells “return” to a progenitor-like state [130,131]. For example, β-cells lacking the TF FoxO1 were found to express markers (i.e., Ngn3) that are specific to endocrine progenitor cells [130]. In accordance with these data, the downregulation of the TFs NKX6-1 and FOXO1 has been shown in human islets from patients with T2D [134]. In a similar fashion, gastrin, a hormone that is only present during embryonic development, was also expressed in T2D mouse models and human diabetic β-cells, highlighting β-cell immature phenotype under diseased conditions [126]. Very surprisingly, in NOD T1D mouse model, an immature surviving β-cell population was also identified. These cells were dysfunctional with reduced granularity, lower expression of mature signature genes (Glut2, ChgA, Pdx1, MafA, Ins2), increased proliferative capacity, higher expression of Ngn3 and increased aldehyde dehydrogenase (Aldh) activity [135]. Furthermore, a surviving β-cell group with similar immature phenotype has been observed in human patients with long standing T1D [136]. These studies suggest that losing the mature phenotype, by proliferation or other dysfunctional mechanisms, might improve the survival of β-cells.

Studies aiming to understand the mechanisms of maintenance of β-cells maturity uncovered a disease mechanism involving Pdx1. In one study, the disruption of the Pdx1 auto-regulatory loop leads to hyperglycemia [137]. In the second, a similar phenotype has been detected by the disturbance of the Pdx1 cooperative function with Foxa2 [138]. Under similar glucotoxic conditions (observed in mouse and human β-cells under diabetic condition), a functional link between GSK3 and Pdx1 was described. Highly active GSK3 phosphorylates Pdx1 causing its degradation, impairing the expression of insulin and Glut2 as well as insulin secretion [139]. Furthermore, Jafaar and colleagues observed in diabetic conditions a metabolic switch, in which β-cells downregulate AMPK to upregulate mTORC1, typically observed in immature β-cells before weaning. Their data also reported that this same switch was recapitulated in samples from patients with T2D [11].

Dedifferentiated β-cell formation, establishment, and function under hyperglycemic conditions (glucotoxicity) is still a matter of study. In this regard, it has been demonstrated that just by removing the hyperglycemic condition these dedifferentiated β-cells reacquire a mature phenotype [126,131]. Along this line, insulin therapy of diabetic mice not only results in the expression of insulin itself together with other maturity markers like Pdx1, MafA and Nkx6-1 [131] but also eliminates gastrin expression [126]. Likewise, implementing fasting mimicking diet (FMD) cycles to diabetic mice reactivate FoxO1 in islets with subsequent expression of Ngn3 and expansion of the β-cell mass [140]. Furthermore, by administration of a GSK3 inhibitor to diabetic islets from T2D human donors, PDX1 functionality and GLUT2 expression were reinstated, making the islets capable to respond appropriately to glucose stimuli and removing glucotoxicity [139]. Thus, by pharmacological or dietary intervention it is possible to halt and ameliorate diabetes by reestablishing the mature β-cell phenotype.

It has been strongly demonstrated that the reduction of β-cell mass is largely caused by apoptosis [136,141,142,143,144]. However, the discovery of β-cell plastic capacity in disease expanded the horizon where β-cell loss of identity under diabetic conditions [34,131,145,146] also contributes to β-cell mass reduction. Evidently, there is a mechanism that enhances β-cell plasticity as a response to the stressful-glucotoxic diabetic environments [130,131], possibly acting as protective. However, understanding whether dedifferentiation is time or severity-dependent process is part of the missing pieces to decipher the full dedifferentiation-plastic mechanism. Finding the key pathways, molecules or endogenous stimuli will provide information for pharmacological treatment that could target and restore dysfunctional β-cells to a functional state.

### 4.2. Polyhormonal Cells and Endcorine Cell Transdifferentiation

Ectopic expression of key TFs and hormones specific for other endocrine cell types [147] indicates the plasticity of β-cells to differentiate toward other endocrine cell types (Figure 3) [148]. Evidence of ectopic expression of other hormones in β-cells comes from a mouse model with impaired insulin secretion and glucose intolerance caused by the lack of Nkx2-2 (TF needed for maintaining mature β-cell identity). Here, β-cells express insulin-somatostatin, insulin-glucagon or even just an alternative hormone and present a polyhormonal feature [149]. The appearance of polyhormonal β-cells together with reduced levels of Ucn3 (maturation marker) in postnatal stages were identified in a hyperglycemic mouse model caused by the interrupted interaction of FoxA2 and Pdx1 TFs [138]. In this regard, Pdx1 has been previously reported to be indispensable for the maintenance of β-cell identity, mainly by repressing the α-cell program [147,150]. Hence, the reduction of Pdx1 activity might result in β-cell transdifferentiation to α-cell fate. Interestingly, in samples from longstanding patients with T1D, non-β endocrine cells show the presence of extremely low, still detectible, levels of insulin and variable levels of the key β-cell markers PDX1, NKX6-1, GLUT1, and PC1/3. Therefore, it seems that in extreme conditions, polyhormonal cells can arise as survival mechanisms and might pinpoint to a “natural” transdifferentiation process [133]. In accordance with the abovementioned work, another study showed the existence of PDX1-expressing α-cells (glucagon positive cells) in biopsies from patients with T1D. The authors speculated that these are exhausted β-cells, transdifferentiated towards the α-cell fate under extreme conditions [151]. These changes in identity might hide the cells from the immune system and prevent the onset or progression of the disease, as shown in another study in which by inducing proliferation and thus maintaining the immature features of β-cells, it was possible to prevent the progression of T1D in NOD mice [152].

The reverse process of transdifferentiation towards β-cells has been an extremely attractive alternative to replenish or restore β-cells, considering that this already happens under completely β-cell mass ablation [147,153]. In a recent elegant study, isolated α-cells from human samples were genetically stimulated for the ectopic expression of the β-cell TFs PDX1 and MAFA. These cells were then transplanted to diabetic mice and not only were they able to produce insulin, but also to reduce hyperglycemia, shedding light on a different regenerative approach [154,155]. Further, in-depth analysis of α- to β-cell transdifferentiation was studied by downregulating two α-cell signature genes, Arx and DNA methyltransferase 1 (DNMT1). With those modifications, α-cells were able to activate the machinery that drives β-cell fate in mice [35]. Transdifferentiation under a challenging condition (diabetes) seems to be more of an attempt to maintain homeostasis in the system. The fact that β-cells have plasticity to the extent of transdifferentiation, opens up opportunities for regenerative therapies. The challenge with transdifferentiation strategies is always in the translatability to clinical models, finding the right combination of TFs and/or uncover the right efficiency and sustainability [155].

### 4.3. Senescence

Recent studies have clarified that β-cells undergo a process of senescence during the physiological aging process (Figure 3). Markers as IGF1R, p16ink4a, p53BP1, and β-galactosidase made possible to observe signs of premature aging and the accumulation of senescent cells in the islets of Langerhans, with reduced functionality if compared to non-senescent cells [156]. The appearance of a premature senescent β-cell subpopulation independent of the aging process has been also found under hyperglycemic and hyperinsulinemic conditions [156,157]. These subpopulations present downregulation of β-cell identity genes (MafA, Nkx6-1, and Pdx1) and upregulation of “disallowed genes” (as Cat and Ldha), together with a senescence-associated secretory phenotype (SASP) [12]. Interestingly, metabolic stress (glucotoxicity) was reported to speed up the appearance of senescent β-cells and loss of β-cell identity [156]. Similarly, a senescent phenotype, before the establishment of hyperglycemia and β-cell death, was observed in islets from NOD mice and patients with T1D [157]. The authors suggest that the inflammatory environment around the islets might be one mechanism at the base of β-cell senescence and SASP. Furthermore, the accumulation of these cells, that fail to be cleared out by the immune system, contributes to the disease progression by promoting activated immune cells infiltration and affecting the neighbor β-cell functionality [157]. Nonetheless, senescent dysfunctional β-cells from insulin resistant mouse models and T1D models (NOD mice) are capable of changing their phenotype into functional and mature upon the exposure to senolytic drugs. Here, β-cells downregulate senescent markers (p21cis1 and p16ink4a) and decrease islet indices of aging and SASP, alleviating the metabolic burden [12]. Overall, this new senescent subpopulation strengthens the idea that there must be a mechanism of β-cell remodeling that is triggered by the environment provided by the diabetic conditions (T1D and T2D). This senescent population was recently found to be present in disease. More needs to be understood in order to know their penetrability and if targeting these cells with senolytic molecules could be a long-lasting solution for diabetes treatment.

In conclusion, the above-mentioned studies highlight 2 key points in β-cell identity: (i) evidence of β-cell plasticity [158] when exposed to a diabetic environment (hyperglycemia, hyperinsulinemia) and ii) their capacity to return to a mature identity and functional phenotype. Thus far, immature, senescent and transdifferentiated β-cell subgroups have been described in diseased conditions. However, there are still some gaps in this area such as i) how healthy and ill β-cells coexist, (ii) how they influence each other in the same islets/pancreas, and iii) what are the mechanisms that determine when β-cells dedifferentiate, transdifferentiate or senesce under disease conditions. Hence, systematic snapshots of known β-cell subpopulations during β-cell maturation and under disease progression will answer some of these questions.

## 5. β-Cell Differentiation Protocols: How to Make “Real” β-Cells

The possibility to differentiate stem cells towards any cell type has opened up an entirely new field for regenerative medicine [159]. The idea to use SC-β-cells for transplantation purposes is fascinating and could potentially help millions of people living with diabetes. Pioneer works from different labs [24,25,26,27] allowed to differentiate iPSCs into β-like cells. Despite the presence of signature genes characterizing the β-cell lineage (PDX1, NKX2-2, NKX6-1, and INS), these protocols have not achieved a functional maturity with a reliable expression of MAFA and human-like dynamic GSIS and in addition a consistent number of poly-hormonal cells are still detectable in the latest stages.

Several recent studies have attempted to improve the differentiation protocols to achieve mature SC-β-cells. For example, Velazco-Cruz and colleagues obtained SC-β-cells with improved dynamic glucose response by i) reducing the differentiating cluster size, ii) manipulating the TGF-β signaling, and iii) depriving the differentiation medium of several components, among which was the serum [64]. Alternatively, Nair and colleagues applied an additional step of aggregation of sorted immature SC-β-cells, in order to improve the functionality of these cells. The authors show that with this step it is possible to emulate some events of the postnatal development that leads to an improved functionality due, among others, to a metabolic maturation of mitochondria [160]. Alagpulinsa and colleagues, instead, focused on the possibility of transplant SC-β-cells together with encapsulation of alginate and CXCL12, which inhibits the fibrotic reaction post-transplant and ensure immune-protection and long-term survival of the graft [161]. Finally, Veres and colleagues clearly mapped, via single cell RNA sequencing (scRNA-seq), each stage of their differentiation protocol, dissecting for each stage of differentiation the milestones to reach islet-like cells [162,163]. Similarly, Alvarez-Dominguez and colleagues have recently mapped the epigenetic landscape of each differentiation step towards SC-β-cells. They unveiled genomic loci fundamental for the α- and β-cell lineage specification and regulatory circuits necessary for the differentiation towards β-cells. Finally, they stretched the importance of the circadian metabolic cycles as critical factors for the achievement of functional mature β-cells [164].

Despite the extensive steps taken forward in the last few years, it is still necessary to improve the current differentiation protocols. Among the most important topics that still need to be clarified, there is the safety of the usage of SC-β-cells in human clinical trials. It will be important to define a clear advanced therapeutic medicinal product with a defined number of SC-β-cells and, similarly, also SC-α-cells that properly function upon transplantation. Furthermore, the time of survival of the graft is still under investigation to evaluate the realistic benefit from the transplantation. In addition, there is a lack of a standardized differentiation protocol that allows in generating α- and β-cells with constant efficiency and minimal percentage of progenitors and poly-hormonal cells, which likely requires sorting and isolation strategies. For the future, we foresee a standardization of the differentiation protocols that aim to obtain constant percentage of monohormonal, fully functional α- and β-cells capable of sensing and responding to glucose variations and shutting down insulin secretion to avoid hypoglycemia. In this regard, animal studies that are currently being conducted to map the physiological propaedeutic steps of β-cell maturation will help to optimize differentiation protocols to generate functional cells for transplantation.

## 6. Concluding Remarks

Over the last decade, major steps in understanding mechanisms underlying β-cell maturation and identity have been taken. Intensive studies in animal models have led to the identification of different molecular markers, functional features and signaling pathways that coordinate β-cell maturation and maintain their functional phenotype. Nevertheless, uncovering how maturation programs are regulated by developmental and physiological parameters such as gene regulatory networks, islet niche factors and diet changes is only at the beginning. Furthermore, there is a lack of understanding of the contribution of the microenvironment, such as islet architecture, innervation, and vascularization in the establishment of mature β-cells. A better understanding of processes that result in maturation of in vitro-derived β-cells as well as their loss of identity in diabetic models might identify common pathways and mechanisms for cell-replacement and regeneration, respectively. In terms of regeneration strategies, more studies are essential to identify the markers and paths that result in loss of β-cell identity for drug targeting to restore functional β-cell mass. In this respect, recent technological breakthroughs such as single-cell genomic and transcriptomic are undoubtedly helping to decipher the pathomechanisms of β-cell failure in patients with diabetes. These advancements together with feasibility of targeted delivery of peptides or small molecules might bring a hope to specifically target and recover the function of those β-cell populations that lost their characteristics.

## Figures and Tables

**Figure 1 ijms-20-05417-f001:**
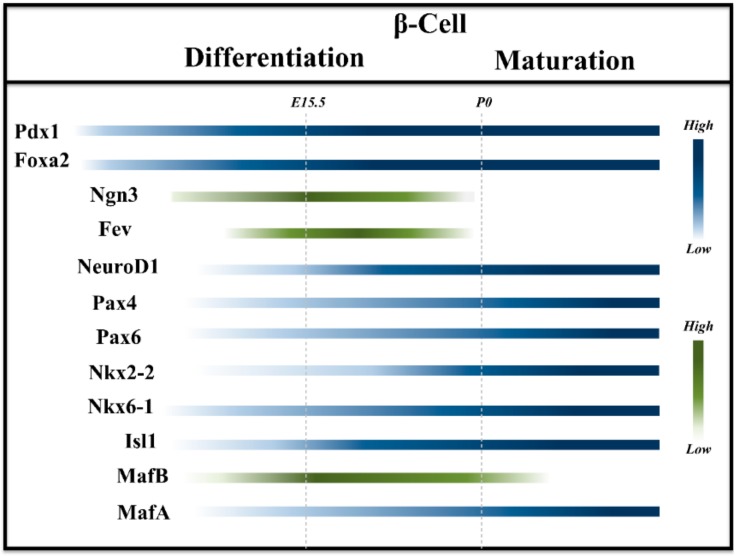
Pivotal TFs regulating β-cell differentiation and maturation in rodents. Schematic representation of the most prominent TFs involved in endocrine induction, lineage specification, differentiation and maturation of β-cells. Gradient colors represent the expression levels during the process. Green colored lines indicate the TFs that are transiently expressed, while the blue ones are those that remain expressed also in adulthood.

**Figure 2 ijms-20-05417-f002:**
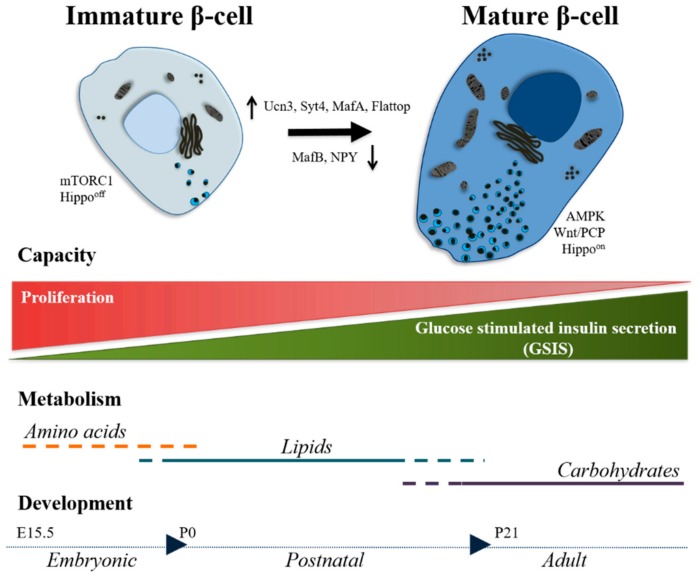
β-cell maturation process. Simplified representation of the complex group of events leading to the β-cell maturation in rodents. In the top part of the figure, β-cells in their immature ad mature stages, characterized by key signaling pathways. Between the two cells, the prominent TFs and markers that are differentially regulated in the maturation process are shown. Below the two most significant capacities are mentioned, proliferation for immature β-cells and GSIS for mature β-cells. Following, the type of metabolisms affecting the maturation stages are mentioned. At the bottom, the timeline of the β-cell maturation process is indicated.

**Figure 3 ijms-20-05417-f003:**
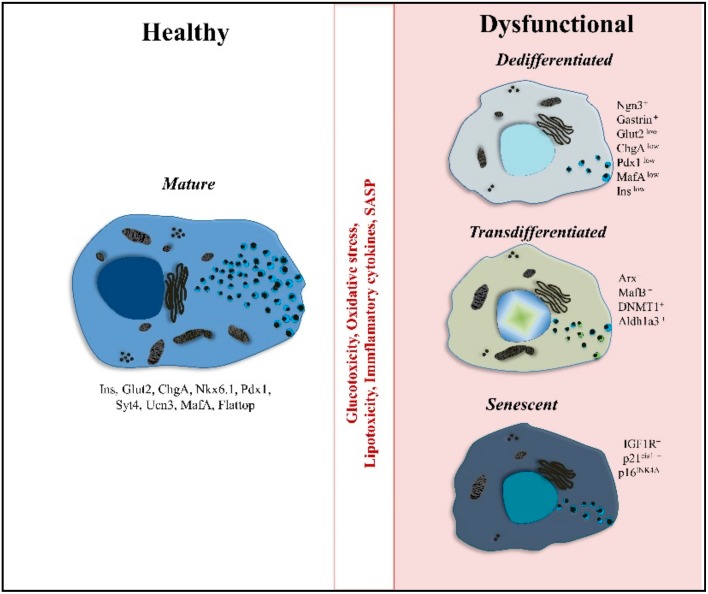
Altered β-cell phenotypes in diabetes. Schematic representation of β-cell described phenotypes as a response to diabetic environments. On the left side, a healthy mature β-cell is shown, in the middle box, disease stressors that induce changes in β-cell phenotype are mentioned, on the right side, summary of the three β-cell phenotypes observed in diabetes: immature/dedifferentiated, transdifferentiated and senescent, with each of the different phenotypes depicting the up- or down-regulation of key markers.

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
