# Peer review of "β-Cell Maturation and Identity in Health and Disease"

_ijms, 2019, doi:10.3390/ijms20215417_

Round 1

Reviewer 1 Report

I would like to thank the authors for this review, it has been a pleasure to read. Overall, I find it is well written, timely and suitable for publication. I only have a few minor suggestions to the authors to improve the manuscript.

Major points
1. A number of references could be added and discussed:

Line 68: It would improve the article if you mentioned in what way the Stem Cell-beta cells are not fully functional. i.e. GSIS is still impaired.

Line 114/125: “Additionally, whether these genes are also expressed in human, and if they have evolutionary 114 conserved function during endocrinogenesis requires further studies.”
It would be interesting to add the study by Hanley lab where they find that the maturation of mouse and human liver are similar while pancreas development is markedly different PMID: 29056335.

Line 125: Unlike rodent, GATA6 is another factor that is required for pancreas formation in humans.
PMID: 22158542

SEction3, first paragraph: please clarify when you are mentioning mouse and / or human.
In many parts of the review, the authors need to be careful to assign the observations/findings to rodents or humans as some studies were exclusively performed in rodents only.
Line 185/Figure 1/2. MAFB is highly expressed in human mature beta cells (almost the same levels as MAFA) and shows the epigenetic signatures of active chromatin. Its specific knockdown in human beta cells results in transcriptional dysregulation, suggesting its roles are not redundant with MAFA in human. PMID: 28041957 (and expression in FACS purified human beta cells: PMID: 23040067)
I would advise to correct text/figures accordingly. Line 307-313: Interestingly exit from cell cycle (EndoC-H3 human cells) also imroves GSIS. Therefore th eimpact of WNT on beta cell function might be indirect, through preventing cell cycle.

Section 4.1: A recent study cited by the authors suggests that the “remaining” beta cells of T1D diabetic patients have a proliferative phenotype. This would be consistent with reduced identity markers (all cycling cells have reduced maturity markers). This could be discussed.

Section 4.2 (382-402) : I would recommend citing evidence that is more clear cut or to severely shorten this section on transdifferentiation of beta cells to poly-hormonal cells. Right now, the evidence cited by the authors look too observational.  Perhaps better to focus on alpha to beta cell differentiation where there is better, clear evidence.  There are minor grammar mistakes throughout. I would recommend an edit by a native-speaker.

Minor points

Line 35: "endemically spread in the world" – please re-phrase. endemic refers to localised/specialised in a region.  "Endemic to earth" is strange as not many studied diabetes outside earth.

Line 48 “Considering diabetes as a reversible disease, alternative therapeutic strategies are being 48 developed” please re-phrase. Only T2D is reversible and this is only in some situations.

Line 52: anti-CD3 does not re-establish beta cell identity. please correct.

line 218: did the authors mean causal, not casual?

Line 228: Thus *reduced* proliferative capacity....

Author Response

Reviewer 1

I would like to thank the authors for this review, it has been a pleasure to read. Overall, I find it is well written, timely and suitable for publication. I only have a few minor suggestions to the authors to improve the manuscript.

We thank the reviewer for their positive feedback on our manuscript.

Major points 
1. A number of references could be added and discussed:

Now, wherever was necessary, we have added the corresponding references.

Line 68: It would improve the article if you mentioned in what way the Stem Cell-beta cells are not fully functional. i.e. GSIS is still impaired. We thank the reviewer for the suggestion.

We have now specified this point in the text (page 2 line 67).

Line 114/125: “Additionally, whether these genes are also expressed in human, and if they have evolutionary conserved function during endocrinogenesis requires further studies.” 
It would be interesting to add the study by Hanley lab where they find that the maturation of mouse and human liver are similar while pancreas development is markedly different PMID: 29056335.
We thank the reviewer for the comment. It is an interesting paper and we have added the paper in the references (Line 115, ref 54) to strengthen the idea that mouse and human pancreas follow different developmental pathways.

Line 125: Unlike rodent, GATA6 is another factor that is required for pancreas formation in humans. 
PMID: 22158542
We thank the reviewer for the suggested article. We have now added the information to the text.

SEction3, first paragraph: please clarify when you are mentioning mouse and / or human. 
In many parts of the review, the authors need to be careful to assign the observations/findings to rodents or humans as some studies were exclusively performed in rodents only. 

We thank the reviewer for the comments. We have tried to specify when the studies are in rodents or human and added to the figure legend that these are visual summaries of the rodents development.

Line 185/Figure 1/2. MAFB is highly expressed in human mature beta cells (almost the same levels as MAFA) and shows the epigenetic signatures of active chromatin. Its specific knockdown in human beta cells results in transcriptional dysregulation, suggesting its roles are not redundant with MAFA in human. PMID: 28041957 (and expression in FACS purified human beta cells: PMID: 23040067)
I would advise to correct text/figures accordingly.

We have also clarified the information regarding MAFB in human, adding also the suggested references.

Line 307-313: Interestingly exit from cell cycle (EndoC-H3 human cells) also improves GSIS. Therefore the impact of WNT on beta cell function might be indirect, through preventing cell cycle.

We have also reinforced the hypothesis that non canonical Wnt might affect the beta cell maturation process via cell-cycle inhibition (line 316)

Section 4.1: A recent study cited by the authors suggests that the “remaining” beta cells of T1D diabetic patients have a proliferative phenotype. This would be consistent with reduced identity markers (all cycling cells have reduced maturity markers). This could be discussed.

We appreciate the reviewer comment and we have strengthen the concept in Line 351.

Section 4.2 (382-402) : I would recommend citing evidence that is more clear cut or to severely shorten this section on transdifferentiation of beta cells to poly-hormonal cells. Right now, the evidence cited by the authors look too observational.  Perhaps better to focus on alpha to beta cell differentiation where there is better, clear evidence.  There are minor grammar mistakes throughout. I would recommend an edit by a native-speaker.

We thank the reviewer for their suggestion. We have now modified the text and start the paragraph with beta cell plasticity and formation of polyhormonal cells. Later we have focused on transdifferentiation of alpha cells into beta cells. 

Minor points

Line 35: "endemically spread in the world" – please re-phrase. endemic refers to localised/specialised in a region.  "Endemic to earth" is strange as not many studied diabetes outside earth.
We thank the reviewer for the comment; we have updated the sentence as “There are two main forms of diabetes”.

Line 48 “Considering diabetes as a reversible disease, alternative therapeutic strategies are being developed” please re-phrase. Only T2D is reversible and this is only in some situations.
We thank the reviewer for the comment; we clearly specify now: “Considering T2D as a possible reversible disease”

Line 52: anti-CD3 does not re-establish beta cell identity. please correct.
We thank the reviewer for the comment. Considering the comments of the Reviewer n.2 we have decided to remove the part regarding the anti-CD3 antibodies.

line 218: did the authors mean causal, not casual?
We thank the reviewer for the comment. We changed the word “casual” for “random”.

Line 228: Thus *reduced* proliferative capacity....
We thank the reviewer for the suggestion; we have added the word “reduced” in the sentence.

Reviewer 2 Report

In their review, Salinno et al. summarize the recent novel knowledge on pathways of b-cell demise in type 1 as well as type 2 diabetes: b-cell dedifferentiation, trans-differentiation and senescence. In their introduction, they provide a good overview on ways towards mature b-cells: their embryonic and postnatal development. The review is timely and well-focused.

I only have some minor comments on its style. In general, English language should be improved.

 1) Abstract: type 1 and type 2 diabetic patients

One should avoid such inappropriate discriminating phrases; change to patients with type 1 and type 2 diabetes throughout the manuscript.

2) Intro:

„In T1D, monoclonal antibodies against CD3 have been tested in order to prevent the onset of the disease by impairing the autoimmune attack against β-cells [7,8]. Similarly, glucotoxicity is the main driver of β-cell loss and identity in T2D [9]. By removing the autoimmune or metabolic stress, by bariatric surgery, diet, anti-CD3 or insulin therapy [10], it is likely possible to improve β-cell function by reestablishing cellular maturation and identity. Therefore, ideally it is possible to protect and regenerate dysfunctional β-cells during disease progression.“

Better rephrase this part, as it is largely not fully accurate:

i) Most clinical CD3 studies have failed.

ii) There is no direct effect by anti-CD3 on the b-cell

iii) Ref 10 is wrong. It is a diet study in T2D

The whole part should be deleted, as obviously not the author’s expertise and such randomly picked therapeutic strategies anyway not focus of this review.

3) Intro line 58: “In the first group”

Which group? Makes no sense

Better change to “strategy”.

The first strategy focuses on (i) finding drugs and small molecules to restore the physiological signaling pathways lost in disease (LIT, e.g. Nat Med. 2014 Apr; 20(4): 385–397), (ii) to remove the dysfunctional cells from the islets (add Lit?) or (iii) to ameliorate the micro-environmental conditions that sustain the impairment of β-cells (add lit, e.g. Alismail, Cell Biosci. 2014; 4: 12).

P.2, line 79: in terms of

P2/74: Numerous reports have shown the loss of β-cell maturation and identity in diabetic conditions.

Add literature

P3/22: on the expression of! Delete „or increased expression levels“

P4/160: UCN3: dont capitalize

P6/222: pregnancy [79]. Add „)“

P6/231-253: GSIS chapter is lengthy and unnecessary here, better shorten to a small single paragraph

P8/305: It is hypothesized...onwards: odd English sentences, please rephrase

P10/372: „The discovery of β-cell plastic capacity in disease allowed to understand that reduction of β-cell mass might not be largely caused by apoptosis, but rather by β-cell loss of identity under diabetic conditions [110,120,130,131].“

I fully agree with the loss of identity part. However, many studies have shown b-cell apoptosis in the past which should not be neglected; intrinsic as well as extrinsic apoptosis machinery activation has been extensively studied and confirmed in well preserved pancreatic sections from patients with T1D as well as T2D.Please rephrase.

Author Response

Reviewer 2

In their review, Salinno et al. summarize the recent novel knowledge on pathways of b-cell demise in type 1 as well as type 2 diabetes: b-cell dedifferentiation, trans-differentiation and senescence. In their introduction, they provide a good overview on ways towards mature b-cells: their embryonic and postnatal development. The review is timely and well-focused.

We thank the reviewer for their positive feedback on our manuscript.

I only have some minor comments on its style. In general, English language should be improved.

 1) Abstract: type 1 and type 2 diabetic patients

One should avoid such inappropriate discriminating phrases; change to patients with type 1 and type 2 diabetes throughout the manuscript.
We thank the reviewer for the comment. We had absolutely no intention to use inappropriate or discriminating sentences. We absolutely understand the power of words and we have decided to change all over the article the sentences “type 1 and type 2 diabetic patients” for “patients with type 1 and type 2 diabetes”.

2) Intro:

„In T1D, monoclonal antibodies against CD3 have been tested in order to prevent the onset of the disease by impairing the autoimmune attack against β-cells [7,8]. Similarly, glucotoxicity is the main driver of β-cell loss and identity in T2D [9]. By removing the autoimmune or metabolic stress, by bariatric surgery, diet, anti-CD3 or insulin therapy [10], it is likely possible to improve β-cell function by reestablishing cellular maturation and identity. Therefore, ideally it is possible to protect and regenerate dysfunctional β-cells during disease progression.“

Better rephrase this part, as it is largely not fully accurate:

i) Most clinical CD3 studies have failed. ii) There is no direct effect by anti-CD3 on the b-cell

iii) Ref 10 is wrong. It is a diet study in T2D

The whole part should be deleted, as obviously not the author’s expertise and such randomly picked therapeutic strategies anyway not focus of this review.
We appreciate the comment of the reviewer. We decided to remove the part on the anti-CD3 antibody, agreeing that the focus of this review is not on therapeutic strategies.

3) Intro line 58: “In the first group”

Which group? Makes no sense

Better change to “strategy”.

The first strategy focuses on (i) finding drugs and small molecules to restore the physiological signaling pathways lost in disease (LIT, e.g. Nat Med. 2014 Apr; 20(4): 385–397), (ii) to remove the dysfunctional cells from the islets (add Lit?) or (iii) to ameliorate the micro-environmental conditions that sustain the impairment of β-cells (add lit, e.g. Alismail, Cell Biosci. 2014; 4: 12).
We thank the reviewer for the comment. We have corrected the entire paragraph and added new references (from line 55 to 64)

P.2, line 79: in terms of
We have corrected the word. We appreciate the comment and thank the reviewer.

P2/74: Numerous reports have shown the loss of β-cell maturation and identity in diabetic conditions.

Add literature
We have added new references (references 35-37)

P3/22: on the expression of! Delete „or increased expression levels“
We have deleted the indicated part.

P4/160: UCN3: dont capitalize
We have wrote the acronym in capital letters, since referred to the human protein.

P6/222: pregnancy [79]. Add „)“
We thank the reviewer for the comment: we have added the parenthesis.

P6/231-253: GSIS chapter is lengthy and unnecessary here, better shorten to a small single paragraph
As the main β-cell feature, we believe it was important to explain the details of the GSIS. We have tried to write more concisely especially in the beginning of the chapter. 

P8/305: It is hypothesized...onwards: odd English sentences, please rephrase
We thank and appreciate the reviewer comment. We have rephrased the entire paragraph.

P10/372: „The discovery of β-cell plastic capacity in disease allowed to understand that reduction of β-cell mass might not be largely caused by apoptosis, but rather by β-cell loss of identity under diabetic conditions [110,120,130,131].“

I fully agree with the loss of identity part. However, many studies have shown b-cell apoptosis in the past which should not be neglected; intrinsic as well as extrinsic apoptosis machinery activation has been extensively studied and confirmed in well preserved pancreatic sections from patients with T1D as well as T2D.Please rephrase.
We agree with the reviewer and we have now strengthen the concept that apoptosis has a major role in the reduction of the beta cell mass, adding multiple references addressing the topic of apoptosis in beta cells during the diabetes onset (lines 376-384).
